# Biodiversity of Actinomycetes from Heavy Metal Contaminated Technosols

**DOI:** 10.3390/microorganisms9081635

**Published:** 2021-07-30

**Authors:** Michaela Cimermanova, Peter Pristas, Maria Piknova

**Affiliations:** 1Institute of Biology and Ecology, P. J. Safarik University, Srobarova 2, 04154 Kosice, Slovakia; michaela.cimermanova@student.upjs.sk (M.C.); peter.pristas@upjs.sk (P.P.); 2Centre of Biosciences, Institute of Animal Physiology, Slovak Academy of Sciences, Soltesovej 4–6, 04001 Kosice, Slovakia

**Keywords:** streptomyces, technosols, heavy metals, biodiversity

## Abstract

Technosols are artificial soils generated by diverse human activities and frequently contain toxic substances resulting from industrial processes. Due to lack of nutrients and extreme physico-chemical properties, they represent environments with limited bacterial colonization. Bacterial populations of technosols are dominated usually by Actinobacteria, including streptomycetes, known as a tremendous source of biotechnologically important molecules. In this study, the biodiversity of streptomycete-like isolates from several technosols, mainly mine soils and wastes (landfills and sludge) in Slovakia, was investigated. The combination of basic morphological and biochemical characterisations, including heavy metal resistance determination, and molecular approaches based on 16S rRNA gene analysis were used for the identification of the bacterial strains. From nine isolates of Actinobacteria collected from different habitats, one was found to represent a new species within the *Crossiella* genus. Eight other isolates were assigned to the genus *Streptomyces*, of which at least one could represent a new bacterial species. Some isolates showed high resistance to Pb, Zn, Cu or Ni. The most tolerated metal was Pb. The results obtained in this study indicate that technosols are a prospective source of new actinomycete species resistant to heavy metals what underlines their bioremediation potential.

## 1. Introduction

Negative impacts on the environment by industry and urbanization represent worldwide problems as a majority of the human population lives near polluted areas. Moreover, contaminants pose a threat to remote areas through emission, leakage into the soil and water and due to accumulation in the food chain. Soils generated or radically transformed by human activities are called technosols or technogenic soils [1]. Lots of the technosols arise as a result of ore mining and ore processing. Such soils are characterized by a high concentration of toxic heavy metals.

Heavy metal ions in living organisms affect cell organelles and components such as the cell membrane, mitochondria, nucleus, DNA and enzymes, causing damage and conformational changes. Production of reactive oxygen species and oxidative stress have been shown to play a key role in the toxicity and carcinogenicity of heavy metals [2]. In addition to being toxic to most organisms in high concentrations, heavy metals are a problem for their stability and non-degradability [3]. Their availability to living organisms is affected especially by soil pH, organic matter, and clay content. These factors affect the formation of the complexes, chelation and the immobilization of heavy metals. Soil acidification has the largest influence due to its strong effects on solubility of metals, which can lead to increased heavy metals bioavailability and reduced prokaryotic enzymatic activity [4,5]. Predominant groups of bacteria in heavy metal-contaminated soil are Proteobacteria, Acidobacteria, and Actinobacteria [6]. To survive adverse conditions, bacteria had to develop resistance mechanisms that involve the transformation of metals into less toxic and less mobile forms by reduction, oxidation, or methylation [7]. Another mechanism is the adsorption of metals on the cell surface, increased efflux pump activity, production of extracellular chelating agents (siderophores), intracellular sequestration and biomineralization [7]. Thanks to these mechanisms, bacteria are used in the remediation of contaminated environments by biological methods called bioremediation [8].

From heavy metal-contaminated areas, several streptomyces strains have been isolated [7]. Genus *Streptomyces* is the most numerous and most common genus of Actinobacteria phylum. Today, this genus consists of almost 700 validly described species [9] and includes Gram-positive, aerobic, filamentous bacteria with a high content of guanine and cytosine in DNA (69–78 mol%). Actinobacteria, especially streptomycetes, are considered the key microorganisms participating in decomposition and recycling of organic matter in soils due to their efficient enzymatic systems. They are known to produce a large number of enzymes having multifaceted industrial applications [10]. Streptomycetes are also important producers of secondary metabolites, producing more than one-third of commercially available antibiotics. Other effects of their metabolites include antimicrobial, antifungal, antiparasitic, antitumor and immunosuppressive activities [10].

At present, the demand for new active substances is growing rapidly, mainly due to the increasing occurrence of resistant pathogens, but also for other needs of medicine, agriculture, and industry. To increase the possibility of the discovery of new species and new bioactive metabolites, research is currently focused on unexplored and harsh environments [11]. Adverse abiotic conditions are thought to select microorganisms capable of adapting to and thus expressing new chemical compounds [12]. The benefit of researching bacteria from contaminated areas lies not only in their use in decontamination and restoration of polluted environments but is also important for the discovery of new chemical structures and new microbial species.

In our work we focused on the isolation and identification of variability of streptomycetes from several technosols in Slovakia. Metal pollution in Slovakia is mainly due to former mining activities, ore processing and industrial plants. In 2019, 103 mining waste repositories were operated, of which 83 were dumps and 20 mining sludges. There are 338 closed and abandoned mining waste repositories registered in the Slovak Republic, of which 28 pose a risk to health and the environment [13]. The most contaminated areas are situated in Central Spiš (Rudňany, Krompachy, and Slovinky municipalities), where excessive contamination of mercury, copper, zinc, arsenic, cadmium, and lead is observed. Other highly contaminated areas are in Jelšava and Hnúšťa municipalities (Hg, Mg, Cd, Pb), Žiar nad Hronom (F, Hg, As) and Dolná Orava (Istebné—Cr, Mn) [14]. In present work we studied streptomycetes from technosol samples from metal-contaminated areas in Slovinky, Gelnica, and Hnúšťa.

## 2. Materials and Methods

### 2.1. Isolation and Characterization of Isolates

For streptomycete isolation, samples from Slovinky, Gelnica, and Hnúšťa mining wastes were used. All sampling areas were formed by remnants from mining activities. Two tailing ponds in Slovinky originated as a result of mining and processing of copper ores. The tailings are near-neutral or slightly alkaline (pH = 7.2–8.8) [15]. Sediments contain high concentrations of Cu, As, Sb, Pb, Zn, and Ba, which are relatively tightly bound in minerals and are characterized by low mobility, mainly due to the slightly alkaline nature of the sediments [16]. Near Gelnica town there are repositories of waste, especially after silver and copper mining. The surrounding soils are characterized by contamination with heavy metals, especially As, Hg and Cu. The content of Zn and Pb also exceeded the limit values [17]. The heap near the town of Hnúšťa was formed as a result of magnesite and talc mining. An increased content of heavy metals, especially As, Pb, Zn and Co, was noted even in the surrounding soils [18].

Substrate samples from sludge ponds and heaps were taken by random sampling to a depth of 10 cm from three locations: Slovinky (two sampling sites, November 2016 [19]), Gelnica (three sampling sites, July 2017) and Hnúšťa (one sampling site, October 2017). Substrate samples were kept at 4 °C and transported to the laboratory as soon as possible. From each technosol, 1 g of sample was shaken in 10 mL of Phosphate Buffered Saline with Tween 20 (PBS-T, Sigma-Aldrich, St. Louis, MO, USA) at room temperature (RT), 130 rpm for 30 min and 100 µL of appropriate dilutions were inoculated on Benedict’s modification of the Lindenbein selective medium for streptomycetes [20]. After incubation at RT for 7 days, colonies with streptomyces-like characteristic were subcultured on Tryptic Soy Agar plates (TSA, Sigma-Aldrich, St. Louis, MO, USA). The isolates were stored as spores’ suspensions in 20% glycerol at −70 °C [21].

Isolates were morphologically characterized by determining the production of diffuse pigments and the color of aerial and substrate mycelium on TSA medium and then classified into a color class [22]. Selected isolates were tested for enzyme activities (proteolytic, lipolytic, amylolytic and cellulolytic) using TSA medium supplemented with milk powder (1% *w*/*v*), glycerol tributyrate (1% *v*/*v*), Starch azure (0.65% *w*/*v*) or carboxymethyl cellulose (2.6% *w*/*v*) [23,24,25]. Growth at different temperatures (RT and 37 °C) was tested on TSA medium and growth at different pH (5 and 7) was tested on TSA medium with 50 mM TRIS (tris(hydroxymethyl)aminomethane) (0.6% *v*/*v*), adjusted with NaOH solution.

### 2.2. Molecular Identification and Diversity Estimation

For DNA isolation, spores (single loop) were inoculated into 50 mL of Tryptic Soy Broth (TSB, Sigma-Aldrich, St. Louis, MO, USA) medium (3 g per 100 mL of TSB) and cultivated aerobically at RT for 2 days at 150 rpm. Total DNA was isolated as described by Nybo et al. with some modifications [26]. Gene for 16S rRNA was amplified by polymerase chain reaction (PCR) using universal bacterial primers fD1 (5′-AGAGTTTGATCCTGGCTCAG-3′) and rP2 (5′-ACGGCTACCTTGTTACGACTT-3′) [27] in a C1000 Thermal Cycler (Bio-Rad Laboratories, Richmond, VA, USA). PCR reactions were performed according to Vandžurová et al. with minor modifications; specifically, we used 25 pmol of each primer (Jena Bioscience, Jena, Germany), denaturation was performed at 95 °C and primer annealing at 54 °C [28]. Amplified fragments were analysed by the restriction fragment length polymorphism (RFLP) method using HaeIII, HhaI, AluI or MspI (Thermo Fisher Scientific, Waltham, MA, USA) restriction endonucleases according to the manufacturer’s instructions. The restriction fragments were separated by electrophoresis in 1.5% (*w*/*v*) agarose gel.

DNA sequencing of selected 16S rRNA amplicons was performed by Eurofins Genomics GATC Services (Eurofins Genomics, Ebersberg, Germany), and the obtained sequences were assembled using MEGA software [29] and compared against the NCBI database using BLASTn algorithm [30]. Accession numbers of sequences deposited to GenBank are MZ438589–MZ438597 for isolates C1, GS1, H6, J1, J2, S2, S7, SL2, and SLA, respectively.

To analyse the phylogenetic relationships among isolates based on 16S rRNA gene sequences, sequences of the most related species were downloaded from the GenBank database, aligned using ClustalW algorithm and phylogenetic tree (Figure 1) was constructed using the neighbor-joining method with 1000 bootstrap replicates using MEGA version X [29]. The species that were heterotypic synonyms [31] were not included in the tree.

### 2.3. Analysis of Heavy Metal Resistance

Determination of resistance to heavy metals of selected isolates was performed by determining the minimum inhibitory concentration (MIC) against zinc (ZnCl_2_), nickel (NiCl_2_), copper (CuCl_2_) and lead (Pb(CH_3_COO)_2_) on modified Duxbury agar [32] supplemented with individual metals. Tested concentrations for Zn, Ni and Cu were 2; 4; 8; 64; 100; 150; 250; 300 mg/L, and for Pb 8; 64; 125; 250; 300; 500; 700; 1000; 1500 mg/L. Inoculated plates were incubated for 7 days at RT. MIC was determined as the lowest concentration with no visible bacterial growth [32]. In order to evaluate the effect of the medium on the growth of isolates in the presence of metal, we tested the growth of isolates at a metal concentration of 64 mg/L on modified Duxbury agar and on complex TSA medium.

## 3. Results

### 3.1. Isolation and Characterization of Isolates

A total of 37 isolates were obtained. Isolated actinomycetes showed similar morphological characteristics and a low rate of melanoid pigment production after 7 days of incubation at RT. Almost all isolates belonged to a white aerial mass color series. The colors of substrate mycelium were mostly beige, yellowish brown to orange. Due to high morphological similarity and low variability of RFLP cleavage profiles (data not shown), we selected for further analysis only 9 different isolates so that each sampling site was represented. Enzyme production was mostly positive. Isolate S7, which hydrolyzed only starch, showed the lowest activity. Overall, we observed low cellulase production, only isolate S2 was characterized by a positive reaction with ratio of the diameter of the clear zone to colony diameter 1.19. Lower pH did not influence the growth of isolates, whereas elevated temperature (37 °C) inhibited the growth of almost half of them. The characteristics of selected isolates are shown in Table 1.

### 3.2. Identification of Isolates

The 16S rRNA sequences of all isolates obtained showed a length of around 1300 bp. BLAST analysis clustered our isolates to six operational taxonomic units (OTUs) at the 99% of identity. 99% sequence identity of full-length sequences is considered an optimal threshold value for species delineation [33,34]. Many assigned species were reclassified by the authors Rong et al., Kim et al. and Liu et al. as heterotypic synonyms [31,35,36,37,38]. Specifically, *Streptomyces coelicolor*, *limosus*, *sampsonii* and *felleus* are a later heterotypic synonym of *S*. *albidoflavus* [35], S. *praecox* is later heterotypic synonym of *S*. *anulatus* [31], *S. albovinaceus*, *griseinus* and *mediolani* is *S. globisporus* [35], the correct name for *S. caviscabies*, *baarnensis*, *acrimycini*, *fimicarius* and *flavofuscus* is *S*. *griseus* [34,36], and for *S. fulvissimus* and *alboviridis* the correct name is *S. microflavus* [34,37]. In the phylogenetic tree (Figure 1), we did not include later heterotypic synonyms, except for *S. coelicolor* (a later synonym of *S. albidoflavus*) and *S*. *flavofuscus* (a later synonym of *S. griseus*) because *S. albidoflavus* and *S. griseus* had slightly lower sequenced identity (99.92%) than their synonyms (100%) to our isolates.

The diversity of sampling sites consisted of five OTUs of streptomycetes and one *Crossiella* sp. (Figure 1). If we do not include heterotypic synonyms but only correct names [9] we can say that the isolate C1 and SLA had 100% sequence identity of the 16S rRNA gene with six different species, namely *S. rubiginosohelvolus*, *S. globisporus*, *S. parvus*, *S. pluricolorescens*, *S. badius* and *S. sindenensis*. Isolate GS1 and SL2 had 100% identity to *S. microflavus*. Isolate H6 showed 100% identity to four species, namely *S. pratensis*, *S. griseus*, *S. anulatus* and *S. cyaneofuscatus*. Isolate S7 had 99.92% identity to the same species as isolate H6 but it forms a separate, well supported, branch in the phylogenetic tree indicating that it could represent a novel species. Isolates J1 and J2 had 100% identity with three species (*S. albidoflavus*, *S. resistomycificus* and *S. griseochromogenes*). S2 isolate was assigned to *Crossiella cryophila* with 16S rRNA gene sequence identity 99.06%. Due to its location in a phylogenetic tree, it may also represent a novel species.

Despite the fact that isolates C1 and SLA as well as GS1 and SL2 shared identical 16S rRNA gene sequences, they probably represent different bacterial species based on their different characteristics (Table 1). Isolates J1 and J2 are more likely to be identical as they were isolated from the same sampling site and share the same characteristics. They differ significantly only in the degree of heavy metal resistance levels.

### 3.3. Heavy Metal Resistance

Heavy metal resistance of selected isolates against Cu, Ni, Pb and Zn on modified Duxbury medium is summarized in Table 2. We noted differences in heavy metal toxicity between isolates even from the same sampling point. All isolates were tolerant to a relatively high concentration of lead in medium (125–1000 mg/L), and the highest MIC for lead was 1000 mg/L in 33% of isolates. According to Duxbury, bacteria can be referred as metal tolerant when can grow in the presence of metal concentration higher than 110 mg/L for Zn, 85 mg/L for Cu and 100 mg/L for Ni [42]; 44% of isolates were tolerant to zinc and copper and 33% to nickel. The highest MIC for Zn, Cu and Ni was 300 mg/L. The most tolerated metal for our isolates was Pb and the most toxic was Cu. The growth of isolates in the presence of metals was more intense on complex TSA medium compared to modified Duxbury agar (Figure 2). Compared to the control (metal-free medium), the percentage growth of isolates on TSA medium was almost 100% for Pb, Cu and Ni and 40% for Zn. On Duxbury agar, isolates growth was 80% for Pb and only about 5% for Zn, Cu and Ni.

## 4. Discussion

The aim of this work was to characterize the isolates from heavy metal-contaminated technosols from several mining and heavy metal industry waste disposal sites in Slovakia (near Gelnica, Slovinky and Hnúšťa). We characterized nine species of which eight belonged to *Streptomyces* genus (closely related to *S. rubiginosohelvolus*, *S. globisporus*, *S. parvus*, *S. pluricolorescens*, *S. badius*, *S. sindenensis*, *S. microflavus*, *S. pratensis*, *S. griseus*, *S. anulatus*, *S. cyaneofuscatus*, *S. albidoflavus*, *S. resistomycificus* and *S. griseochromogenes*). The last isolate was found to belong to the *Crossiella* genus with the highest similarity to *Crossiella cryophila*. *Streptomyces* spp. have already been characterized from technosols from heavy metal industry waste disposal sites in Slovakia, including those in Sereď and Žiar nad Hronom [43]. The technosols from a nickel sludge disposal site near Sereď are characterized by strongly alkaline pH and high content of heavy metals, especially Cr and Ni [44,45]. Actinobacteria were found to be a dominant phylum of cultivable bacteria with *Arthrobacter* as a dominant genus. *S. variabilis* was identified as a single representative of *Streptomyces* genus [43]. In earlier study, *Actinomyces* spp. were also isolated from farmland near waste disposal site of nickel smelter near Sereď and among other isolates two *Streptomyces* spp. closely related to *S.*
*collinus* and *S.*
*exfoliatus* were isolated [46]. From brown mud disposal site from aluminum production near Žiar nad Hronom, two streptomycete species closely related to *S. wuyuanensis* and *S. malaysiensis* (based on 16S rRNA sequences) were isolated [43,47]. Site is characterized by elevated concentrations of heavy metals like As, Pb, Hg and Cr. *Streptomyces* K11 isolate showed resistance to zinc up to 150 mM and high zinc bioaccumulation capacity [47].

To date, several heavy metal-resistant streptomycetes have been isolated from various contaminated areas [7]. Álvarez et al. analysed the spread of resistance to heavy metals in the phylogeny of streptomycetes based on bibliographic data and sequence availability for the 16S rRNA gene, confirming the broad spread of resistance in different branches [7]. In his work, they summarized heavy metal resistant strains of streptomycetes from accessible bibliographies. In Table 3, we depict novel heavy metal-resistant species which were isolated from metal-contaminated areas. The literature review (Appendix A: List of strains from heavy metal-contaminated areas) suggests that many *Streptomyces* species with unique characteristics were isolated from soils of metal-contaminated areas [32,47,48,49,50,51,52,53,54,55,56,57,58,59,60,61,62,63,64,65,66,67,68,69,70,71,72,73,74]. For example, Schmidt et al. isolated streptomycetes highly resistant to nickel from a polluted site at the former uranium mine [57]. Similarly, El Baz et al. isolated from various abandoned mining areas located around Marrakech heavy metal-resistant *Streptomyces* sp., capable of accumulating lead [32]. Isolate *Streptomyces* sp. Z38 from rhizosphere soil contaminated with pesticides and heavy metals characterized as *S. griseorubens* was able to grow abundantly in the presence of 250 mg/L of Cr(IV) and produced plant growth-promoting metabolites [63,64]. Isolate *Streptomyces* sp. H-KF8 with antimicrobial activity against *S. aureus*, *L. monocytogenes*, and *E. coli* was isolated from marine sediments with the natural occurrence of heavy metals due to volcanic activity. It was also characterized by the tolerance to various heavy metals, such as Cu, Co, Hg, Cr, Ni and Te, with 49 genes likely involved in heavy metal resistance [65,66]. The high resistance to various heavy metals and the ability of bioaccumulation and biosorption indicates the possibility of use of isolates in the bioremediation of contaminated areas [32,75]. Moreover, adaptation to such adverse conditions may cause the production of new secondary metabolites [12]. Adverse environments are, therefore, the focus of interest in the search for new species of Actinobacteria, antibiotics and other useful substances for medicine and the economy [76].

This work, like many previous ones, points to low variability within the 16S rRNA gene for species identification of streptomycetes. However, it allows us to estimate groups of closely related species and to classify isolates into the clades or OTUs [77]. Labeda et al. [77] studied almost all described species of the Streptomycetaceae family (615 species at the time) based on the sequences of the 16S rRNA gene. They found that this family contained 130 statistically based clades as well as many unsupported and single member clusters. Our isolates were related mainly to species not described by Labeda et al. or to clade 112 (*S. albidoflavus*). El Baz et al. [32] isolated also some species related to clade 112. Other species from reviewed literature were mainly from OTUs not reported by Labeda et al. or from clade 103 (*S. tendae*). The observed high species variability of streptomycete populations from heavy metal-contaminated areas indicates that different autochthonous species are able to adapt to high concentrations of metals. It is well known that taxonomic characterization of *Streptomyces* is complicated and challenging because species sharing identical small subunit rRNA sequence can exhibit clearly different properties and secondary metabolism, as reported e.g., by Antony-Babu et al. [78]. Therefore, additional phylogenetic markers, such as MLSA analysis of housekeeping genes, with discriminatory power better than 16S rRNA gene, are essential to evaluate the evolutionary relationships among closely related *Streptomyces* species [10,31]. This underlines the importance of examining the properties of specific strains, in order to discover their true potential.

The identification of *Crossiella* sp. in heavy metal-contaminated technosols is quite surprising. The type strain of the genus, *Crossiella cryophila* was isolated from pristine soil from Shosenkyo, Yamanashi Prefecture, Japan and to date, two species of the genus, *C. cryophila* and *C. equi* have been described [9]. So far, there have only been a few records of this species in the literature. Our isolate, despite related to *C. cryophila*, showed significant heavy metal resistance and differed from the type strain [79] in several biochemical properties. Due to its position in the phylogenetic tree, it may represent a new species of this small genus.

Regarding the resistance of our isolates to heavy metals, most of our isolates showed high resistance to selected heavy metals on modified Duxbury agar. Maximum concentrations at which our isolates were able to grow were 250 mg/L (3.82 mM) for Zn, 250 mg/L (3.93 mM) for Cu, 250 mg/L (4.26 mM) for Ni, and 700 mg/L (3.38 mM) for Pb. Most resistant isolates were GS1 and SL2, closely related to *S. microflavus*. However, comparing resistance with other available literature is relatively difficult as the authors use different solid and liquid media which have, as confirmed also in our work, a significant impact on the bioavailability of metals for microorganisms [32,56]. We can say that the composition of the medium has a significant effect on the toxicity of metals and Duxbury agar affects the bioavailability of metals less than TSA medium (Figure 2), as also stated by El Baz et al. [32]. Lead was one of the most tolerated heavy metals in our isolates and similar results were obtained by El Baz et al. [32]. More important than the resistance itself is the application of isolates for bioremediation of polluted areas. Sometimes even strains with lower tolerance to heavy metals can be better metal adsorbents than highly resistant ones. This was observed e.g., for lead tolerance in *S. viridochromogenes*, when the most adsorbing strain was at the same time the most sensitive one to this metal [52]. The usability of isolates in bioremediation and their biosynthetic potential will be the subject of further study.

## 5. Conclusions

Extreme environments, including metal-contaminated areas, are generally underexplored. However, it has become clear that they present a promising source of novel microorganisms with the genus *Streptomyces* being undoubtedly the most prominent producer of putative new bioactive molecules and with a bioremediation potential. Our study explores the diversity of streptomycete-like isolates with high metal resistance from heavy metal-contaminated technosols in Slovakia. We identified nine isolates of which eight belonged to *Streptomyces* genus, closely related to several *Streptomyces* species. One isolate belonged to the *Crossiella* genus. Some of our isolates were highly resistant to metals with the highest MIC for Pb 1000 mg/L and for Zn, Cu and Ni 300 mg/L. Some isolates exhibited notable enzymatic activity, mainly proteolytic and amylolytic. The results obtained indicate that at least two of our isolates could be representatives of new bacterial species within *Streptomyces* (isolate S7) and *Crossiella* genera (isolate S2).

## Figures and Tables

**Figure 1 microorganisms-09-01635-f001:**
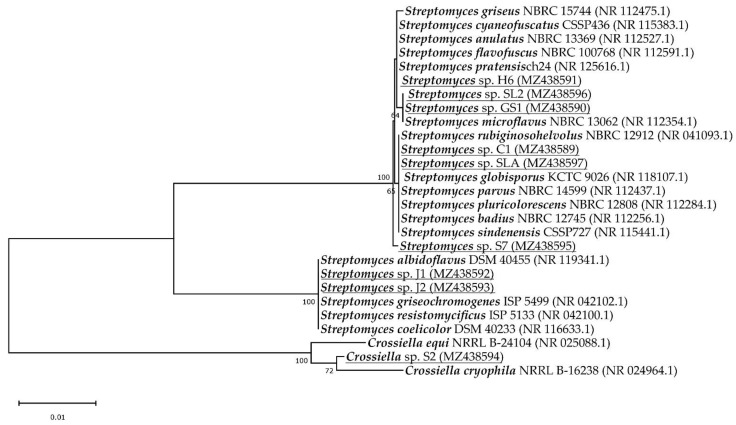
The phylogenetic tree based on the 16S rRNA gene sequences. The evolutionary history was inferred using the neighbor-joining method [39]. The optimal tree with the sum of branch length = 0.13126272 is shown. The percentage of replicate trees in which the associated taxa clustered together in the bootstrap test (1000 replicates) are shown next to the branches [40]. The tree is drawn to scale, with branch lengths in the same units as those of the evolutionary distances used to infer the phylogenetic tree. The evolutionary distances were computed using the Kimura 2-parameter method [41] and are in units of the number of base substitutions per site. The analysis involved 27 nucleotide sequences, the isolates included in this study are underlined. All ambiguous positions were removed for each sequence pair (pairwise deletion option). There was a total of 1274 positions in the final dataset. Evolutionary analyses were conducted in MEGA X [29].

**Figure 2 microorganisms-09-01635-f002:**
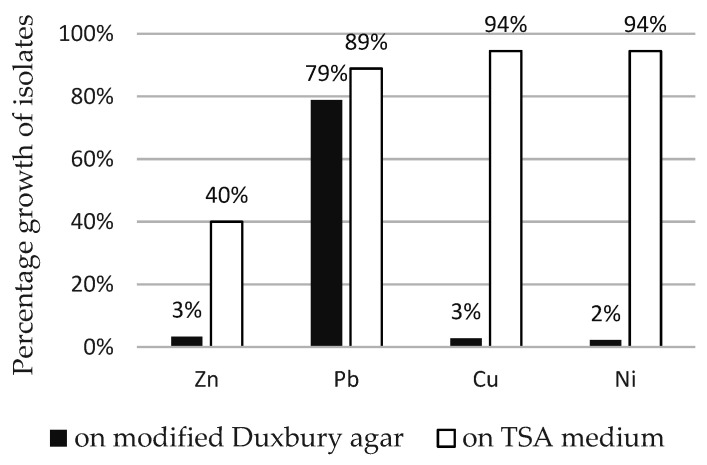
Percentage growth of isolates on modified Duxbury agar medium and Tryptic Soy Agar (TSA) medium with Zn, Pb, Cu and Ni at concentration 64 mg/L against control (media without metal).

**Table 1 microorganisms-09-01635-t001:** Morphological characteristics of selected isolates, enzyme production and growth characteristic.

Sampling Site	Isolate	Substrate Mycelium	Aerial Mycelium	Pigment	Protease	Lipase	Amylase	Cellulase	37 °C	pH 5	pH 7
Gelnica	C1	O	W	-	+	±	+	±	+	+	+
	GS1	B	W	-	+	-	+	±	-	+	+
	H6	B	W	Br	+	+	+	±	-	+	+
	J1	B	W	-	+	+	+	-	+	+	+
	J2	B	W	-	+	+	+	-	+	+	+
Hnúšťa	S2	B	W	-	-	±	-	+	-	+	+
	S7	O	C	Br	-	-	+	-	-	+	+
Slovinky	SL2	B	W	-	+	±	+	±	+	+	+
	SLA	Y-Br	W	-	+	±	+	±	+	+	+

B: beige, Br: brown, C: colorless, O: orange, W: white, Y-Br: yellow-brown, “+” positive reaction/growth, “±” slightly positive reaction, “-” negative reaction/no pigment/no growth.

**Table 2 microorganisms-09-01635-t002:** Minimal inhibitory concentration (MIC) of isolates on modified Duxbury agar.

Sampling Site	Isolate	MIC [mg/L]			
		Cu	Ni	Pb	Zn
Gelnica	C1	4	8	300	250
	GS1	300	300	1000	300
	H6	300	8	1000	300
	J1	2	100	300	100
	J2	2	8	300	8
Hnúšťa	S2	64	100	250	100
	S7	300	150	125	64
Slovinky	SL2	300	250	1000	300
	SLA	4	8	300	64

**Table 3 microorganisms-09-01635-t003:** Novel *Streptomyces* species isolated from heavy metal contaminated areas from reviewed literature.

Isolate	Strain	Acc. Number	Place of Isolation	Country	Reference
*S. cadmiisoli*	ZFG47	NR_171522.1	soil in a cadmium-contaminated area in Xiangtan City	Hunan Province, China	[71]
*S. cyaneochromogenes*	MK-45	NR_170501.1	manganese-contaminated area	Xiangtan, China	[68]
*S. manganisoli*	MK 44	KY911452.1	manganese-polluted soil, Xiangtan Manganese Mine	South Central China	[70]
*S. plumbiresistens*	CCNWHX 13-160	EU526954	lead-polluted soil, Gansu province	Northwest China	[60]
*S. sporoverrucosus*	dwc-3	KC508633.1	disposal site for (ultra-)low uraniferous radioactive waste	Southwest China	[67]
*S. xiangtanensis*	LUSFXJ	NR_164877.1	manganese-polluted soil, Xiangtan Manganese Mine	South Central China	[72]
*S. zinciresistens*	CCNWNQ 0016	GU225938	zinc and copper mine, Shaanxi province	Northwest China	[61]

## Data Availability

All 16S rRNA data obtained through this study were deposited to the GenBank database.

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
