# Peer review of "Biodiversity of Actinomycetes from Heavy Metal Contaminated Technosols"

_microorganisms, 2021, doi:10.3390/microorganisms9081635_

Round 1

Reviewer 1 Report

The manuscript deals with the isolation and identification of variability of streptomycetes from several technosols in Slovakia  The subject of the manuscript is consistent with the scope of the Journal. This manuscript provided some insightful information on the technosols in the context of prospective source of new streptomycete species with unique properties. Authors received many interesting results which were correctly interpreted. They discussed the results in an interesting way against the background of numerous items of world literature.  I recommend this manuscript to publication in this journal but after small corrections. Detailed comments are provided below.

Abstract

It is not clear from the text whether it is a review publication or own research. The reader has the impression that it will be a review. Authors should rewrite the abstract in terms of their research (issue, short description, results, conclusion).

 „Our data indicate that technosols are a prospective source of new streptomycete species with unique properties”. „…unique properties”?? – explain.

Introduction

 „Their availability to living organisms is affected by soil pH, organic matter, and clay content [4]”. - Explain how availability depends on these properties

The authors should describe the role of actinomycetes in the environment, referring, for example, to the ability to decompose organic matter. After all, the authors studied these ability.

Materials and methods

Explain all abbreviations used the first time (RT, TSA, TSB)

TSA, TBS - specify the composition, company name

„PCR reactions were performed according to Vandžurová et al. with minor modifications [26]”. - specify exactly what the modifications were

Results

„Overall, we observed low cellulase production” -  What about the highest activity?

“We can say that the composition of the medium has a significant effect on the toxicity of metals and Duxbury agar affects the bioavailability of metals less than TSA medium, as stated by El Baz et al. [29].” - Take that sentence into the discussion and refer to your results

Discussion

„Sometimes even strains with lower tolerance to heavy metals can be better metal adsorbents than highly resistant ones [49].” - expand on this statement from which it may be due.

Conclussion

Conclusions are too general. Authors should include specific conclusions from their research and their relevance

Reviewer 2 Report

Presented manuscript describes classical microbiological research on the biodiversity of streptomycetes in unusual ecological niches - technosols. The overall merit on this manuscript is completely positive: interesting results were obtained and presented in good form. However, I have several minor comments to make this manuscript more clear. 

  1. I'd recommend to change the Title a bit: due to the results obtained, in my opinion it would be better to title this manuscript "Heavy metal contaminated technosols as a source of resistant actinomycetes species". Writing "unique" in the title creates some misunderstanding on nouvel species isolation.
  2. line 11 - Actinobacteria should be started with capital letter.
  3. Sentence on lines 36-38 - reference on this mechanism should be added.
  4. line 46 URL address should be formatted as reference [№]
  5. Lines 62-81 - I'd recommend to rewrite this paragraph and move information about the mines and it's history to the materials and methods section.
  6. Line 88 - PBS-T should be clarified: phosphate saline buffer with tween 20 (PBS-T)
  7. line 98 - room temperature should be written full at first mention.
  8. line 115 - the same situation with URL-link format
  9. lines 116-117 Accession numbers could be presented like MZ438589 - MZ438597 for isolates  ... , respectively. 
  10. line 167 & 289 - URL-link format
  11. table 3 - I'd recommend to title the last column as "reference"
  12. References section should be formatted according to the journals' requirements
